# The effects of prolonged sitting, prolonged standing, and activity breaks on vascular function, and postprandial glucose and insulin responses: A randomised crossover trial

Meredith C. Peddie[1]*, Chris Kessell[2], Tom Bergen[2], Travis D. Gibbons[2], Holly A. Campbell[3], James D. Cotter[2], Nancy J. Rehrer[2], Kate N. Thomas[3]

1 Department of Human Nutrition, University of Otago, Dunedin, New Zealand, 2 School of Physical Education, Sport and Exercise Sciences, University of Otago, Dunedin, New Zealand, 3 Department of Surgical Sciences, Dunedin School of Medicine, University of Otago, Dunedin, New Zealand

* meredith.peddie@otago.ac.nz

## Abstract

The objective of this study was to compare acute effects of prolonged sitting, prolonged standing and sitting interrupted with regular activity breaks on vascular function and postprandial glucose metabolism. In a randomized cross-over trial, 18 adults completed: 1. Prolonged Sitting; 2. Prolonged Standing and 3. Sitting with 2-min walking (5 km/h, 10% incline) every 30 min (Regular Activity Breaks). Flow mediated dilation (FMD) was measured in the popliteal artery at baseline and 6 h. Popliteal artery hemodynamics, and postprandial plasma glucose and insulin were measured over 6 h. Neither raw nor allometrically-scaled FMD showed an intervention effect (p = 0.285 and 0.159 respectively). Compared to Prolonged Sitting, Regular Activity Breaks increased blood flow (overall effect of intervention p<0.001; difference = 80%; 95% CI 34 to 125%; p = 0.001) and net shear rate (overall effect of intervention p<0.001; difference = 72%; 95% CI 30 to 114%; p = 0.001) at 60 min. These differences were then maintained for the entire 6 h. Prolonged Standing increased blood flow at 60 min only (overall effect of intervention p<0.001; difference = 62%; 95% CI 28 to 97%; p = 0.001). Regular Activity Breaks decreased insulin incremental area under the curve (iAUC) when compared to both Prolonged Sitting (overall effect of intervention P = 0.001; difference = 28%; 95% CI 14 to 38%; p<0.01) and Prolonged Standing (difference = 19%; 95% CI 4 to 32%, p = 0.015). There was no intervention effect on glucose iAUC or total AUC (p = 0.254 and 0.450, respectively). In normal-weight participants, Regular Activity Breaks induce increases in blood flow, shear stress and improvements in postprandial metabolism that are associated with beneficial adaptations. Physical activity and sedentary behaviour messages should perhaps focus more on the importance of frequent movement rather than simply replacing sitting with standing.

**Data Availability Statement:** All relevant data are within the manuscript and its Supporting Information files

**Funding:** This research was generously supported by a grant from: University of Otago. MCP was supported by a fellowship from The National Heart Foundation of New Zealand (grant no.1518), with support from the Southland Medical Foundation. None of these funders had any role in study design, data collection and analysis, decision to publish or preparation of the manuscript.

**Competing interests:** The authors have declared that no competing interests exist.

## Introduction

Sedentary behaviour (performing activities when sitting or lying that involve < 1.5 MET of energy usage) is associated with increased risk of mortality, cardiovascular disease and type 2 diabetes [1, 2], but regular high levels (e.g. >60 min) of moderate-to-vigorous physical activity may prevent such associations [3]. Most intervention studies designed to test the causality of the association between sedentary behaviour and health outcomes have focused on the *acute* effects of regularly interrupting periods of prolonged sitting with short bouts of activity on postprandial metabolism. A growing number of studies have now been conducted in this area, confirming that ~1.5 min– 5 min of light or moderate activity performed every 20–30 min can improve postprandial glucose metabolism when compared to 2–9 h of uninterrupted sitting in participants ranging from obese type 2 diabetics to healthy inactive young adults [4–7]. An alternative method for reducing periods of prolonged sitting is to encourage standing. However, the results of the small number of acute studies conducted indicate that short bouts of standing [8], or even alternating 30 min of standing with 30 min of sitting [9], are not as effective at reducing postprandial glycemia as regular *activity* breaks, particularly in healthy, normal weight populations [8]. Very little is known about the effects of more prolonged bouts of standing, despite recommendations being made to encourage individuals to stand for at least half of their work day [10]. Impaired endothelial function precedes the development of atherosclerosis [11] and is predictive of cardiovascular events [12, 13]. Changes in circumferential pressure and shear stress (the forces associated with blood flow) play an important role in the prevention or development of endothelial dysfunction and atherosclerosis [14]. Prolonged sitting reduces blood flow to the lower limbs, decreasing shear stress (the tangential force of the flowing blood on the endothelial surface), and resulting in transient endothelial dysfunction [15]. Additionally, a greater volume of sedentary behaviour has also been associated with chronically impaired endothelial function [16, 17]. It is likely that reductions in blood flow and endothelial function also contribute to the elevated postprandial glycemia observed with prolonged sitting given that muscle provides the largest sink for glucose disposal, and that glucose uptake is facilitated by muscle perfusion [18]. Reduced blood flow to the large muscles of the legs thereby reduces the major role those muscles can play in the regulation of glucose metabolism. Findings from the few studies published to date indicate that regular activity breaks [15, 19, 20], standing [21] or standing breaks [22] may protect against the negative impact that sitting has on endothelial function. However, all studies on normal weight individuals were of sitting in a fasted state [15, 21] or after a light or low-energy meal [19, 20]. Feeding can affect vascular function, and therefore, these studies do not represent the postprandial state most common during waking hours. The aim of this study was to compare the effects of prolonged sitting, prolonged standing and sitting interrupted with regular activity breaks on endothelial function and postprandial glycemic response.

## Materials and methods

### Study design

This randomized, controlled, cross-over trial took place at the University of Otago, in Dunedin, New Zealand between April 2019 and July 2019. The trial was approved by the University of Otago Human Ethics Committee (Health), approval number: H18/138, and written informed consent was obtained from all participants before screening. The study is registered prospectively with the Australian New Zealand Clinical Trial Registry (ANZCTR12619000175178).

## Participants

Participants were recruited through the distribution of emails and advertisements placed on campus-related social media sites. Eligible participants were 18–40 y of age who self-reported sitting for >5 h per day, did not smoke, were free of cardiovascular disease or diabetes, had a BMI <30 kg/m$^2$, were not on medication known to influence endothelial function or postprandial metabolism, were capable of participating in physical activity and standing for long periods (assessed via administration of PAR-Q and self-report), and had a blood pressure at screening of <140/90 mm Hg (Fig 1).

## Experimental protocol

The experimental protocol is presented in Fig 2. Participants completed three 6-h interventions: Prolonged Sitting, in which participants sat for 6 h; Prolonged Standing, in which participants stood at a standing desk for 6 h; and Regular Activity Breaks, in which sitting was interrupted by 2 min of walking on a treadmill (5 km·h$^{-1}$; 10% incline) every 30 min. The sequence in which participants completed the three interventions was random (see below). Each intervention was separated by a minimum of 4 d (median 7 d; 25$^{th}$ and 75$^{th}$ percentiles 6 and 20 d, respectively). The only time participants deviated from their prescribed posture during each intervention was during two predetermined bathroom breaks. At each bathroom break participants walked slowly ~20 m down the hall to the bathroom (regardless of whether then needed to use the facilities) and back again. When sitting or standing, participants were permitted to read or use their laptop (for work or leisure). An investigator was in the room with participants throughout each session to ensure adherence to all protocols.

## Randomization

Participants were randomly assigned to complete the three interventions in one of the six possible orders. The randomization sequence was generated prior to participant recruitment by MCP using Stata software (version 15 for MAC; StataCorp, College Station, Texas), and concealed electronically. The afternoon prior to each participant beginning their first intervention session the next sequential randomization sequence was revealed and assigned.

## Standardization of prior diet and exercise

Participants were asked to avoid physical activity for 24 h prior to each intervention session, and record everything they ate and drank. A copy of this record was returned to participants prior to each subsequent intervention and they replicated the food and drink consumed as closely as possible. Participants wore an ActiGraph accelerometer (ActiGraph GT3X+, ActiGraph, Pensacola, Florida) and an interstitial glucose monitor (Freestyle Libre Pro, Abbot Diabetes Care) over this time to monitor compliance.

## Meals

All participants consumed the same breakfast (muesli, trim milk, juice, toast, margarine and honey) at baseline, and a snack (chocolate brownie) at four hours. The breakfast and snack combined provided 4697 kJ energy, 170 g carbohydrate (62% energy) and 35 g fat (28% energy). Gluten-free alternatives were provided on request (n = 3) and provided 4716 kJ energy, 180 g carbohydrate and 33 g fat. All food was consumed within 15 min by all participants, and no other food was consumed over this time. Water intake was provided ad libitum during the first session and then replicated in all subsequent sessions.

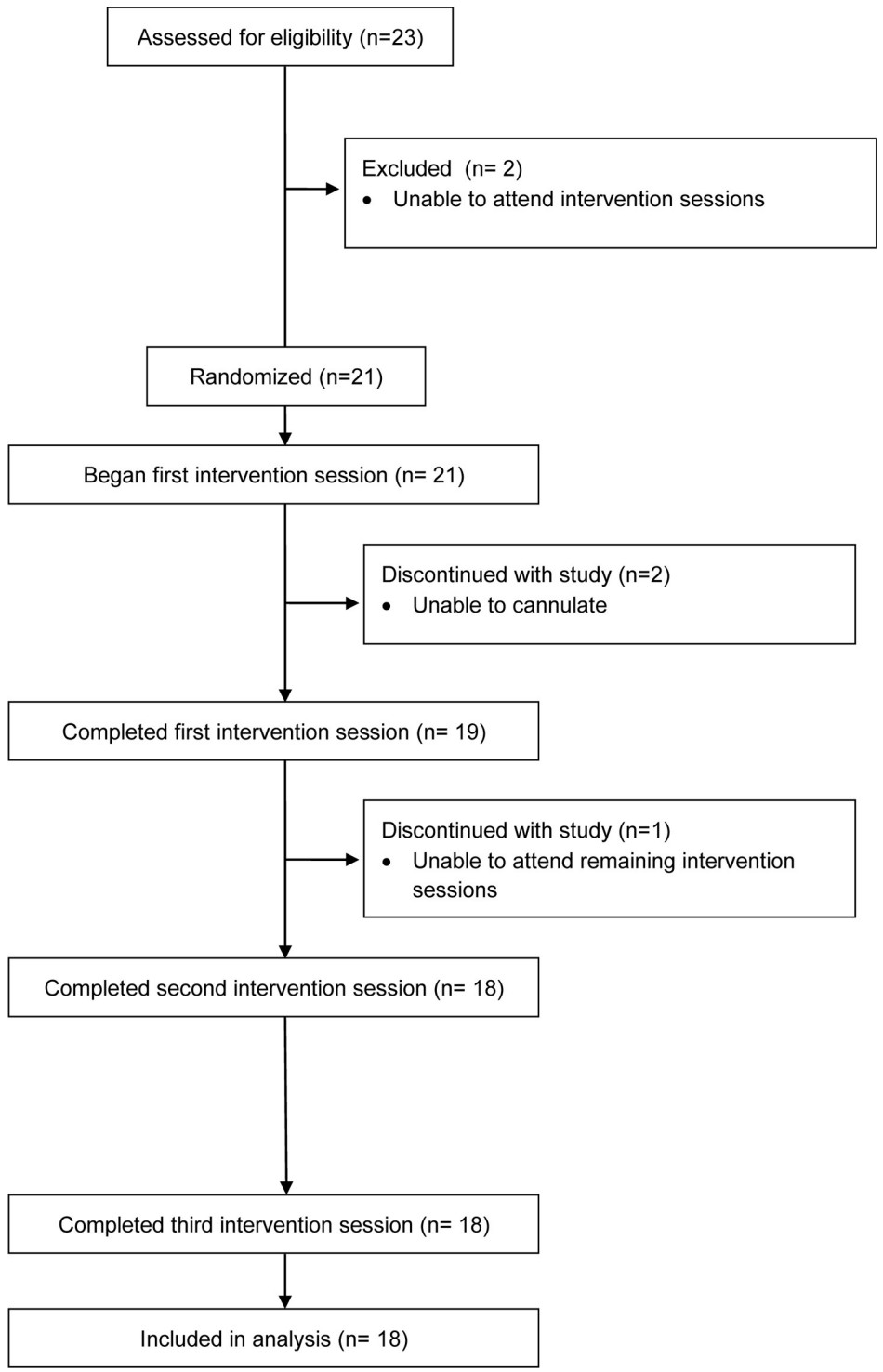

**Fig 1. Participant flow diagram (CONSORT).**

## Measurement of popliteal artery hemodynamics

The popliteal artery was assessed by one of two experienced sonographers (KNT and TDG), and all measurements within individual participants were completed by the same sonographer.

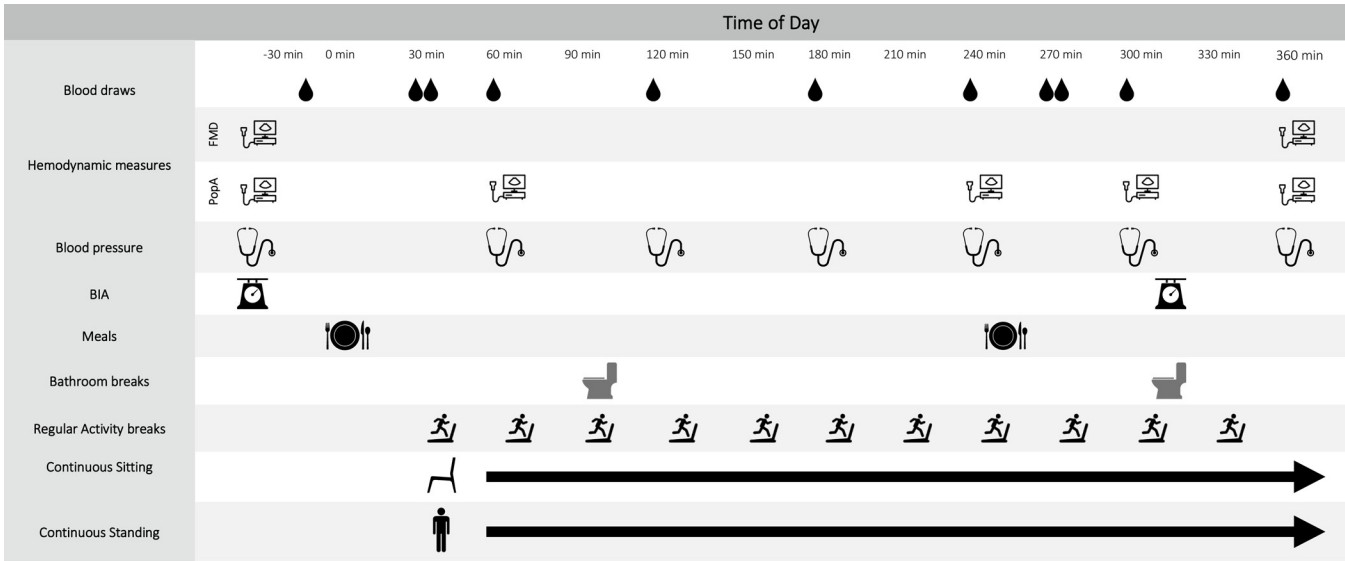

**Fig 2. Intervention session timeline.**

Simultaneous popliteal artery diameter and blood velocity were measured using ultrasound (Terason uSmart 3300, MA, USA) with a 15 MHz linear array transducer (bandwidth 4–15 MHz). Hemodynamics were assessed for ~60 s on each occasion. All baseline and final (360 min) measurements were made with participants in a left lateral recumbent position with the knee slightly flexed, as were all measurements during Sitting and Regular Activity Breaks interventions. However, in the Prolonged Standing intervention, hourly measures taken between 0 and 360 min were in a standing position with the knee slightly flexed. In the Regular Activity Breaks intervention measurements were taken immediately prior to an activity break, meaning the participant had been sitting continuously for ~30 min prior to measurement. The position of measurement within the popliteal fossa was marked to ensure consistency of location. This location was measured from bony landmarks and/or the knee skin crease and replicated in subsequent sessions. The location was verified by comparing the recording from the baseline of the previous intervention session prior to the first assessment in a subsequent intervention. Ultrasound depth, focus position, gain and Doppler settings were optimized for each participant for the first measurement and replicated thereafter. Videos were recorded using Camtasia Studio Screen Recording Software (TechSmith, MI, USA). Offline analysis was performed blinded to intervention assignment, using wall-tracking software (Cardiovascular Suite v 3.6, Quipu, Pisa, Italy).

## Measurement of Flow Mediated Dilation (FMD)

Flow mediated dilation was measured in the popliteal fossa following published guidelines [23] at baseline and 360 min, after the participant had rested quietly for >10 min in a left lateral recumbent position. Following a 2-min baseline, transient ischemia was induced by inflating a 12-cm cuff around the mid calf to 200 mm Hg within 2 s (SC12 cuff, TD312 blood pressure cuff inflator; Hokanson, Bellevue WA, USA). Occlusion was maintained for 5 min and recording continued for 3 min following rapid release of the cuff. Baseline diameter ($D_{base}$), blood flow velocity (v) and shear rate (SR) were assessed as the mean of the last minute of recording prior to cuff inflation. Blood flow was calculated as the product of half the time-averaged peak envelope velocity and vessel cross-sectional area. Peak diameter post deflation

was determined automatically using edge-detection software. Flow-mediated dilation was calculated as the percentage increase in diameter from baseline (FMD = $(D_{peak} - D_{base}) / D_{base}$ x 100), following guidelines utilising allometric scaling to adjust for $D_{base}$ with a covariate-controlled approach [24].

## Muscle mass of left lower limb

Muscle mass of the left lower limb was assessed via bioelectrical impedance (InBody 230, InBody, Seoul, South Korea) at baseline and immediately prior to the bathroom break at 315 min. Changes in muscle mass between the two measurements were assumed to be changes in fluid in the lower limb.

## Blood pressure

Arterial blood pressure was measured hourly using an automated sphygmomanometer (OMRON HEM-907) with an appropriately-sized cuff. In all intervention sessions, at baseline (0 min) and 360 min blood pressure was measured in a seated upright position after at least 10 min of sitting. In the Prolonged Sitting and Regular Activity Breaks interventions blood pressure was also measured in a seated upright position for all other measurements. In the Regular Activity Breaks intervention blood pressure measurements were conducted immediately prior to an activity breaks meaning the participant had been sitting for ~30 min prior to each measurement. In the Prolonged Standing intervention blood pressure was measured in a standing position at all other time points. Measurements were taken on the same arm for all conditions, contralateral to the arm with the indwelling venous cannula. Three measurements were taken one minute apart and the mean of the three measurements recorded at each time point.

## Blood collection

Participants arrived at the laboratory in a fasted state. Immediately after baseline FMD measurements were completed a cannula was inserted into a vein in the antecubital fossa or forearm. At least 10 min later a fasting blood sample was collected and the session began. Blood samples were collected hourly for 6 h, with additional samples collected 30 and 45 min after both breakfast and the snack; i.e., 11 samples were collected in total. Blood samples were collected into a syringe and transferred immediately into EDTA-containing tubes. At baseline, 180 and 360 min a capillary tube was filled with blood collected from the syringe for analysis of hematocrit. Both the capillary and EDTA containing tubes were stored chilled until they were centrifuged (1100 rpm for 5 min and 300 rpm for 10 min, respectively), within 2 h of collection. Hematocrit was measured using a custom-made Vernier calliper reader immediately after centrifugation, while plasma from the EDTA tube was aliquoted and stored at -80˚C for later analysis.

## Analytical methods

Plasma glucose concentration was measured using the hexokinase enzymatic method on a Cobas C 311 analyser (Roche Diagnostics; Mannheim, Germany). Plasma insulin concentration was measured using electrochemiluminescence methods on an Elecsys 2010 (Roche Diagnostics; Mannheim, Germany). Kits and calibrators for glucose and insulin analysis were sourced from Roche Diagnostics (Mannheim, Germany). All samples from the same participant were assayed in a single run. Intraassay CVs were 0.7% for glucose and 1.3% for insulin. All analyses were performed by a lab technician who was blinded to intervention allocations.

## Statistical analysis

Data were analysed using Stata software (version 16.0 for PC; StataCorp, College Station Texas). The primary outcome for this study was the difference in change in FMD from baseline to 360 min between interventions. A sample size of 18 participants provided 80% power to detect a 0.85 reduction in $D_{base}$-adjusted FMD ($\alpha = 0.05$) assuming a sample size of 1.0, using a test of paired comparisons, and allowing three participants to complete the three interventions in each of the 6 possible intervention orders. This sample size also allowed 80% power to detect a 10% difference in glucose area under the curve (AUC) and a 15% difference in insulin AUC. The effect of the different interventions on raw and allometrically scaled FMD at 230 min was evaluated using linear mixed models with adjustment for baseline diameter.

The effect of the different interventions on the hemodynamic variables (diameter, shear rate and blood flow) was assessed at three time points of interest using linear mixed models with adjustment for values measured at 0 min: 1) at 60 min (to assess the very acute effects of the posture/activity); 2) the mean of the measurements taken at 240 and 300 min (to assess the posture/activity after it had been sustained for several hours while incorporating the effects of a second feeding) and; 3) at 360 min (to assess if changes as a result of the posture/activity were maintained after returning to a seated position). For ease of interpretation, differences in flow between interventions were converted to percentages post-analysis.

Mean blood pressures for each intervention were quantified as the mean of all time points *after* baseline. The effect of the different interventions on mean systolic and diastolic blood pressure was evaluated using mixed model regression.

The total AUCs for glucose and insulin were calculated using concentrations measured from each of the 11 time points, using the command *integ* in Stata. This calculation fits a cubic spline curve through the time points on which blood samples were collected and calculates the area under that curve. Incremental AUC (iAUC) were also calculated by subtracting baseline concentrations from each subsequent concentration over the 6 h period and calculating the area under the curve using the same command as above. When samples were missing due to cannula malfunction (n = 4) the area under the curve was calculated without the missing time point (no baseline or 360 min samples were missing). The effect of different interventions on total AUC and iAUC for glucose, and insulin was evaluated using linear mixed models. For ease of interpretation, differences in AUC between interventions are presented as percentages which were calculated post-analysis.

For all mixed models, period and order affects were assessed individually and found not to have an important effect, however, for completeness order was included as a predictor in all final models. The assumptions of regression models were checked for every model and found to be met for all models other than insulin (in which the residuals were skewed). Log transformation addressed the non-normality to a large extent, but robust standard errors were used to allow for small deviations from normality. For consistency, robust standard errors were then used in all models. No adjustment was made for multiple comparisons.

## Results

Participant flow through the study is presented in Fig 1. Eighteen participants completed all three interventions sessions and were included in the final analysis. Participant characteristics are summarized in Table 1. Two thirds of participants (12/18) reported habitually participating in five or more hours of physical activity a week. There were no differences between interventions in mean (sd) interstitial glucose (Prolonged Sitting: 5.5 (0.6) mmol·L$^{-1}$; Prolonged Standing: 5.5 (0.4) mmol·L$^{-1}$; Regular Activity Breaks: 5.3 (0.5) mmol·L$^{-1}$; p = 0.120) or the mean total number of activity counts (sd) (Prolonged Sitting: 526,802 (330,383) counts; Prolonged

**Table 1. Participant characteristics.**

| Characteristic | All (n = 18) |
|---|---|
| Females [n (%)] | 7 (39) |
| Age (y) | 23.5 (5.0) |
| Height (cm) | 175.0 (10.2) |
| Weight (kg) | 72.9 (12.1) |
| BMI (kg·m$^{-1}$) | 23.7 (2.6) |
| Systolic blood pressure (mm Hg) | 116 (9) |
| Diastolic blood pressure (mm Hg) | 73 (12) |
| Fasting glucose (mmol·L$^{-1}$) | 4.9 (0.5) |
| Fasting Insulin (pmol·L$^{-1}$) | 49.6 (33.8) |

Data presented as mean (SD) unless otherwise stated

Standing: 626,154 (308,260) counts; Regular Activity Breaks: 495,945 (313,010) counts: p = 0.666) measured in the 24 h prior to participants arriving at the laboratory.

There was no effect of intervention on either raw or allometrically scaled FMD at 360 min (after controlling for FMD at 0 min) (p = 0.285 and 0.159 respectively, Table 2).

There was significant effect of intervention on net shear rate at 60 min (p<0.001), the mean of the 240 and 300 min measurements (p<0.001) and at 360 min (p = 0.001) (Fig 3). At 60 min Regular Activity Breaks resulted in a net shear rate that was 72% higher (95% CI 30 to 114%; p = 0.001) than Prolonged Sitting (after controlling for net shear rate at 0 min). This difference was maintained between 240 and 300 min (difference = 62%; 95% CI 30 to 89%; p<0.001) and at 360 min (difference = 83%; 95% CI 45 to 120%; p<0.001) (Fig 3). These changes appear to be driven by increases in antegrade shear rate during Regular Activity Breaks. There was a significant effect of intervention on antegrade shear rate at 60 min (p = 0.02), the mean of the 240 and 300 min measurements (p<0.001) and at 360 min (p<0.001) At 60 min, Prolonged Standing resulted in a net shear rate that was 80% higher (95% CI 38 to 122%; p<0.001) than Prolonged Sitting. This appeared to be mediated by a decrease in retrograde shear rate associated with Prolonged Standing (overall intervention effects for retrograde shear rate were significant at 60 min (p<0.001), the mean of the 240 and 300 min measurements (p<0.001) but not at 360 min (p = 0.199)). This difference dissipated between 240 and 300 min (difference 39%;

**Table 2. Flow Mediated Dilation (FMD) responses measured during 360 min of prolonged sitting, prolonged standing and regular activity breaks in 18 healthy, normal weight adults.**

| Time | | Baseline | 360 min | | |
|---|---|---|---|---|---|
| | | | mean | Δ from baseline | p$^{*}$ |
| **Raw FMD (%)** | | | | | |
| | Prolonged Sitting | 2.5 (1.6) | 3.5 (2.6) | 0.8 (2.4) | 0.285 |
| | Prolonged Standing | 3.1 (2.3) | 2.9 (2.0) | -0.3 (2.3) | |
| | Regular Activity Breaks | 3.6 (2.3) | 2.6 (3.0) | -0.8 (2.62) | |
| **Allometrically scaled FMD (%)** | | | | | |
| | Prolonged Sitting | 2.4 (1.4) | 3.5 (2.1) | 1.04 (1.03 to 1.05)** | 0.159 |
| | Prolonged Standing | 3.0 (2.1) | 2.6 (2.0) | 0.99 (0.98 to 1.01)** | |
| | Regular Activity Breaks | 3.7 (2.3) | 3.0 (2.7) | 0.99 (0.98 to 1.05)** | |

$^{*}$p for overall effect of intervention at 360 min, with adjustment for baseline values

** presented as ratio of geometric means (95% CI)

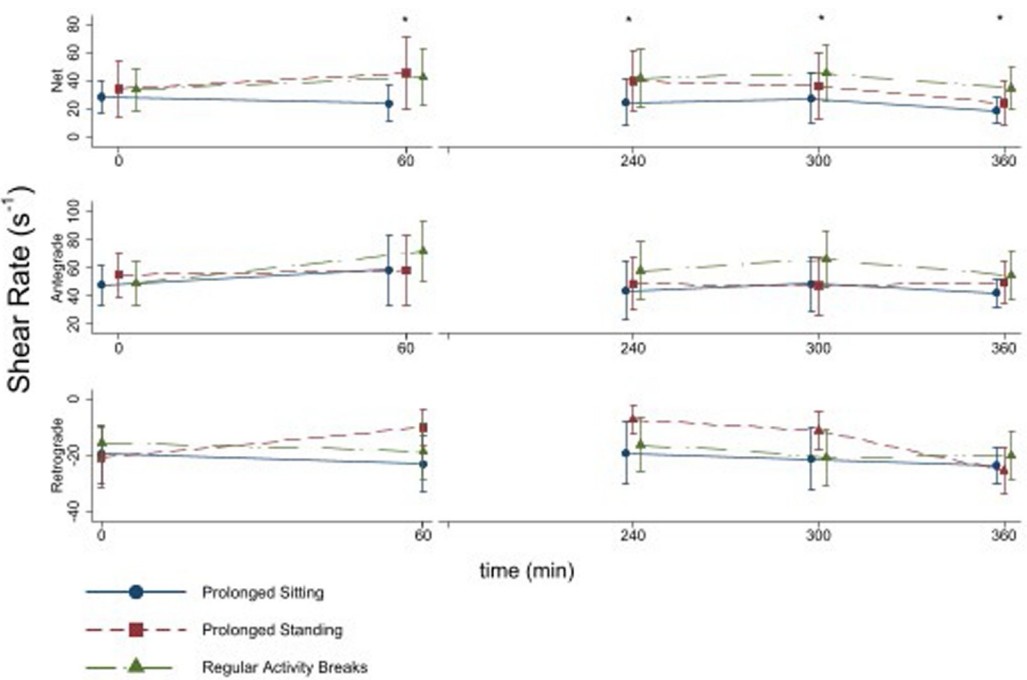

**Fig 3. Mean total antegrade and retrograde shear rate measured over the 6-h intervention period in n = 18 healthy participants.** Error bars are standard deviations. All baseline (0 min) and final (360 min) measurements were made with the participants in a left lateral recumbent position, as were all measurement in the Prolonged Sitting and Regular Activity Breaks interventions. In the Prolonged Standing intervention measurements made at 60, 240 and 300 min were performed with the participant standing with the knee slightly flexed.

95% CI -4 to 81; p = 0.080), and was not apparent at 360 min (difference 21%; 95% CI -21 to 64%; p = 0.329). Net shear rate was not different between Regular Activity Breaks and Prolonged Standing at 60 min (difference -4%; 95% CI -33 to 57; p = 0.768), or between 240 and 300 min (difference 19%; 95% CI -13 to 48%; p = 0.257), however, at 360 min Regular Activity Breaks resulted in a net shear rate that was 46% higher (95% CI 4 to 87%; p = 0.026) than Prolonged Standing (after controlling for net shear rate at 0 min) (Fig 3). There was no effect of intervention on diameter at any measurement time (Table 3).

There was a significant effect of intervention on blood flow at 60 min (p<0.001), between 240 and 300 min (p<0.001) and at 360 min (p<0.001) (Table 3). At 60 min Regular Activity Breaks resulted in a blood flow that was 80% larger than that seen with Prolonged Sitting (95% CI 34 to 125%; p = 0.001). This difference was maintained between 240 and 300 min (difference = 71%; 95% CI 39 to 104%; p<0.001) and 360 min (difference = 110%; 95% CI 54 to 156%; p<0.001). Prolonged Standing increased blood flow at 60 min by 62% (95% CI 28 to 97%; p = 0.001) when compared to Prolonged Sitting. However, this difference was no longer apparent between 240 and 300 min or at 360 min (Table 3). Blood flow was not different between Regular Activity Breaks and Prolonged Standing at 60 min (difference = 11%; 95% CI -23 to 44%; p = 0.549) or between 240 and 300 min (difference = 32%; 95% CI -2 to 64%; p = 0.069). However, at 360 min Regular Activity Breaks increased blood flow by 56% when compared to Prolonged Standing (95% CI 10 to 105%; p = 0.019).

Prolonged Standing increased the measured muscle mass of the lower limb (an indirect marker of fluid accumulation; overall effect of intervention p<0.001) at 315 min by 0.4 kg (95% CI 0.2 to 0.5 kg; p<0.001) when compared to Prolonged Sitting, and by 0.3 kg (95% CI 0.1 to 0.4 kg; p = 0.008) when compared to Regular Activity Breaks. However, the difference at

**Table 3. Popliteal artery diameter and blood flow responses measured during 360 min of prolonged sitting, prolonged standing and regular activity breaks in 18 healthy, normal weight adults.**

| Time | Baseline | 60 min | | | 240–300 min | | | 360 min | | |
|---|---|---|---|---|---|---|---|---|---|---|
| | | mean | % Δ from baseline | p* | mean | % Δ from baseline | p* | mean | % Δ from baseline | p* |
| **Diameter (mm)** | | | | | | | | | | |
| Prolonged Sitting | 5.9 (0.8) | 5.9 (0.9) | 2 (5) | 0.229 | 5.8 (0.8) | 2 (7) | 0.300 | 6.0 (0.9) | 2 (7) | 0.325 |
| Prolonged Standing | 5.8 (0.8) | 5.7 (0.8) | -2 (9) | | 5.7 (0.6) | 0 (7) | | 5.9 (0.7) | 0 (7) | |
| Regular Activity Breaks | 5.9 (0.8) | 6.0 (0.7) | 2 (10) | | 5.9 (0.5) | 3 (10) | | 6.1 (0.6) | 3 (10) | |
| **Blood flow (mL·min⁻¹)** | | | | | | | | | | |
| Prolonged Sitting | 36.2 (23.1) | 28.9 (18.4) | -20 (72) | P<0.001 | 30.8 (24.2) | -15 (78) | P<0.001 | 22.4 (8.9) | -38 (53) | P<0.001 |
| Prolonged Standing | 38.8 (24.3) | 47.6 (26.3) | 25 (84) | | 40.8 (22.3) | 6 (86) | | 30.5 (35.3) | -21 (75) | |
| Regular Activity Breaks | 40.8 (23.3) | 52.3 (22.9) | 28 (65) | | 54.3 (29.1) | 33 (58) | | 47.8 (25.7) | 16 (56) | |

*p for overall effect of intervention at specified time point, with adjustment for baseline values

315 min was not meaningfully different between Prolonged Sitting and Regular Activity Breaks (difference 0.1 kg; 95% CI -0.1 to 0.4; p = 0.238)

The overall effect of intervention was not significant for systolic blood pressure (p = 0.654), however, it was for diastolic blood pressure (p = 0.010) (Table 4). Specifically, Prolonged Standing increased mean diastolic blood pressure by 8 mm Hg (95% CI 2 to 10 mm Hg; p = 0.006) when compared to Prolonged Sitting, and by 5 mm Hg (95% CI 1 to 8 mm Hg; p = 0.004) when compared to Regular Activity Breaks. Diastolic blood pressure did not differ significantly between Prolonged Sitting and Regular Activity Breaks (1 mm Hg; 95% CI -1 to 4; p = 0.347 (Table 4)).

When adjusted for baseline concentrations there was no significant difference (p = 0.713) in hematocrit at 180 min (mean (SD) hematocrit = 43 (4) for Prolonged Sitting; 44 (5) for Prolonged Standing and 44 (4) for Regular Activity Breaks) or at 360 min (p = 0.166; mean (SD) hematocrit = 43 (4) for Prolonged Sitting; 43 (4) for Prolonged Standing and 43 (3) for Regular Activity Breaks).

The plasma glucose and insulin concentrations measured over the 6-h intervention are shown in Fig 4, while the total and iAUC are shown in Table 4. The overall effect of intervention was not significant for either total or incremental AUC for glucose (p = 0.254 and 0.450, respectively), but was for total and incremental AUC for insulin (p = 0.001 and 0.005, respectively; Fig 4). Regular Activity Breaks decreased insulin total AUC by 27.5% (95% CI 13.6 to 37.6%; p<0.01) when compared to Prolonged Sitting, and by 19.3% (95% CI 4.1 to 32.1%, p = 0.015) when compared to Prolonged Standing. The effect of the Prolonged Sitting and Prolonged Standing interventions on insulin total AUC did not differ significantly (8.9%; 95% CI -5.1 to 21.8%; p = 0.224). Similarly, Regular Activity Breaks decreased insulin iAUC by 44.5% (95% CI 20.8 to 61.1%; p = 0.01) when compared to Prolonged Sitting, and by 36.0% (95% CI 11.0 to 54.0%, p = 0.008) when compared to Prolonged Standing. The effect of the Prolonged Sitting and Prolonged Standing interventions on insulin total AUC did not differ significantly (13.3%; 95% CI -3.2 to 27.2%; p = 0.109).

## Discussion

This is the first study to measure the time course of the vascular and metabolic responses to Prolonged Sitting, Prolonged Standing and Regular Activity Breaks, and after consumption of

**Table 4. Summary responses over 360 min of prolonged sitting, prolonged sitting and regular activity breaks measured in 18 healthy, normal weight adults.**

| | Mean (SD) | p* |
|---|---|---|
| **Systolic blood pressure (mmHg)** | | |
| Prolonged Sitting | 110.9 (7.7) | 0.654 |
| Prolonged Standing | 109.9 (8.2) | |
| Regular Activity Breaks | 109.9 (8.2) | |
| **Diastolic blood pressure (mmHg)** | | |
| Prolonged Sitting | 62.4 (6.9) | 0.010 |
| Prolonged Standing | 68.1 (8.5) | |
| Regular Activity Breaks | 63.8 (6.5) | |
| **Total glucose AUC (mmol·L$^{-1}$·360 min)** | | |
| Prolonged Sitting | 1890.2 (238.5) | 0.254 |
| Prolonged Standing | 1816.6 (148.8) | |
| Regular Activity Breaks | 1831.1 (193.1) | |
| **Incremental glucose AUC (mmol·L$^{-1}$·360 min)** | | |
| Prolonged Sitting | 189.6 (142.0) | 0.450 |
| Prolonged Standing | 152.3 (150.6) | |
| Regular Activity Breaks | 140.9 (136.7) | |
| **Total insulin AUC (pmol·L$^{-1}$·360 min)** | | |
| Prolonged Sitting | 69365.9 (1.3)** | 0.001 |
| Prolonged Standing | 63202.6 (1.4)** | |
| Regular Activity Breaks | 50875.2 (1.5)** | |
| **Incremental insulin AUC (pmol·L$^{-1}$·360 min)** | | |
| Prolonged Sitting | 58788.5 (50987.8 to 67783.0)** | 0.005 |
| Prolonged Standing | 52904.8 (41519.4 to 67412.1)** | |
| Regular Activity Breaks | 33352.1 (19511.7 to 57010.1)** | |

*p for overall effect of intervention
**presented as geometric mean (95% CI)

a high-carbohydrate meal. Performing Regular Activity Breaks increased both blood flow and net shear rate in the popliteal artery when compared to Prolonged Sitting, within 60 min of starting this pattern of activity, and then maintained this increase over the 6 h period (with the measurement of blood flow occurring ~30 min after each activity break in the seated position). This increase in blood flow and shear rate remained after 10 min in the left lateral recumbent posture prior to the measurement at 360 min. Standing initially increased blood flow to similar levels as seen with regular activity breaks at 60 min, but this was not maintained between 240 and 300 or 360 min. Meaningful changes in FMD were not observed in response to any intervention.

## Hemodynamic effects

Few studies have examined vascular effects of prolonged sitting and/or activity breaks. This, combined with differences in methodology between studies, make comparisons difficult. Other studies have examined from 90 min to 6 h of sitting, using a variety of 'breaking' interventions (e.g., standing, fidgeting, short walks, longer walks, callisthenics), and in various arteries (superficial femoral, popliteal or brachial). Furthermore, the current study was performed with participants in a fed state. On balance, most studies have demonstrated reduced blood flow and net shear rate during and following sitting for 3–6 h, and most interventions

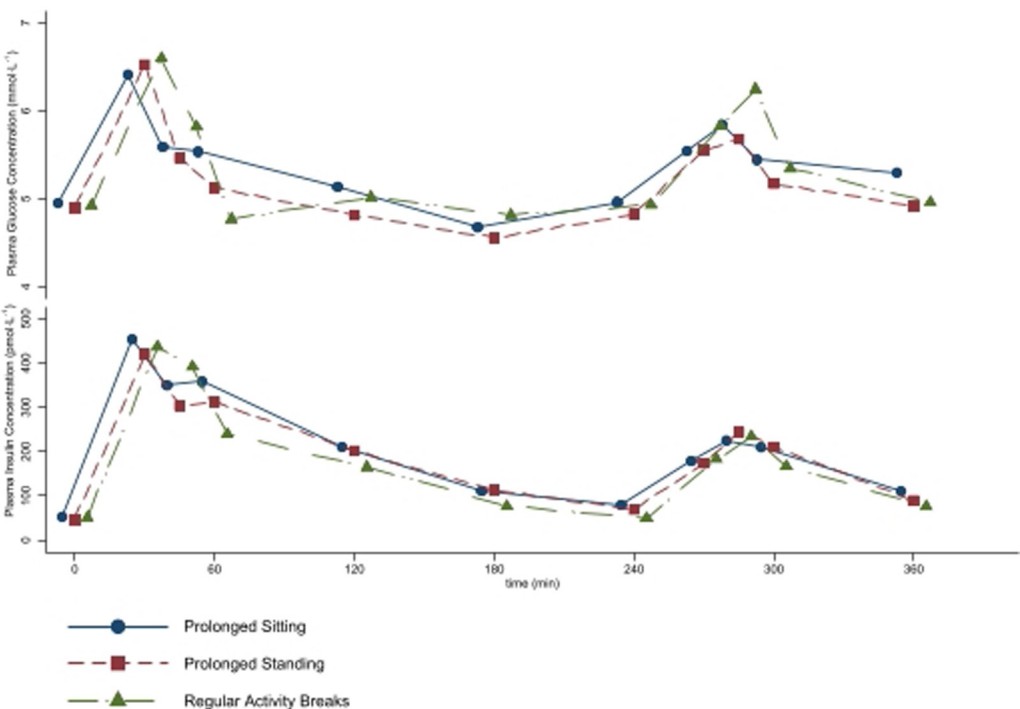

**Fig 4. Mean plasma glucose and insulin concentrations measured over the 6-h intervention period in n = 18 healthy participants.** Meals were fed at 0 and 240 min.

that interrupt sitting mitigate these reductions [15, 19, 25–27]. The results of the current study support that blood flow and net shear rate are lower with prolonged sitting, and that activity breaks ameliorate this response. Blood flow and shear rates in the popliteal artery during walking are also going to be significantly higher, so the measurement ~30 min post-walking underestimates the extent of the favourable hemodynamic stimulus. However, the finding of elevated blood flow between walking bouts is important for understanding the effect that walking has *during* the seated periods. Standing appears to prevent the decline in blood flow and shear rate initially, but this protection is not maintained past 4 h; a time effect that is perhaps caused by the slow development of edema in the leg that opposes the effects of gravity on arterial blood pressure. The absence of this fluid shift seen with Regular Activity Breaks may contribute to the elevation in blood flow and shear rate observed for the entirety of the 6-h periods. The 6-h standing period was longer than used in previous studies; for example, Morishima et al. [21] demonstrated increased blood flow and shear rate throughout 3 h of standing.

Despite these relatively consistent hemodynamic profiles across sitting / breaking studies, effects on FMD are more variable, with a number of studies [15, 21, 25, 26] documenting a reduction in FMD with sitting and protection by breaking, and others showing no change following any intervention [19, 20, 22]. It is unclear as to why the 6-h sitting intervention did not reduce FMD in the current study. Perhaps in this young, otherwise healthy, homogeneous population, the stimulus was insufficient to evoke an impairment, or the walk to the toilet on two occasions (performed by every participant regardless of need)was enough to mitigate the effects of sitting for long periods. However, from the observations of reduced blood flow and shear stress, it is possible that over time, i.e., repeated exposures, this might precipitate endothelial dysfunction.

## Metabolic effects

Performing regular activity breaks reduced postprandial insulin response when compared to both prolonged sitting and prolonged standing. The reduction in insulin response when compared to prolonged sitting is in line with previously summarised literature [8]. However, the fact that prolonged standing did not seem to offer any benefit to glucose metabolism when compared to prolonged sitting provides support to the idea that standing is less likely to result in beneficial metabolic effects in healthy participants [28]. The lack of reduction in postprandial glucose is in contrast to the results of several other studies [8], however, many of the earlier studies provided very high levels of carbohydrate. It is possible that–particularly in healthy individuals–it is harder to observe reductions in glucose response when meals are closer in composition to what would be consumed in real life. However, it is important to keep in mind that while the glucose response is not significantly different, the insulin response is reduced with regular activity breaks indicating an acute improvement in insulin sensitivity. It seems likely that the maintained increase in blood flow in the lower limb seen with regular activity breaks is helping to facilitate increases in non-insulin mediated glucose uptake [29, 30], which if maintained with habitual behaviour may have important implications on overall cardio-metabolic health.

The results of the current study indicate that even healthy individuals benefit from interrupting sedentary time with brisk walking. Perhaps instead of providing standing desks to individuals who sit for prolonged periods at work, we should be redesigning the physical and psychosocial occupational landscapes to provide employees with opportunities to regular interrupt periods of prolonged sitting with short bouts of physical activity [31].

## Limitations and delimitations

Consideration should be given to the following aspects of the study design that may have influenced the results. Our study assessed acute responses to a 6-h period of sitting, standing and performing regular activity breaks. While this duration of standing is one of the longer assessed in this context, the chronic impact of these patterns of activity on vascular function and glycemia response are yet to be fully elucidated. The study included only young, healthy participants. There has been suggestion that participants who are less metabolically healthy may have larger positive responses to interventions that interrupt prolonged sitting [28], and therefore, the magnitude of the responses reported here may underestimate the response in the population as a whole. We did not adjust for sex in our analysis, nor did we control for menstrual cycle. While the cross-over design somewhat eliminated the need to adjust for sex, non-adjustment or control for menstrual phase may have introduced variability [32], future studies should consider evaluating any effects of menstrual cycle phase. We used changes in muscle mass measured via bioelectrical impedance to represent fluid shifts in the lower limb. Future studies should consider more direct measurements of limb edema and consider assessing leg symptoms. We did not adjust for multiple comparisons in the statistical analysis. While some arguments support this approach [33], small differences between interventions should be interpreted with caution. Leg movement was not controlled across any of the interventions, nor was it quantified so that it could be included in statistical modelling. It is possible that if we limited fidgeting of the lower limb, or at least adjusted for it statistically that this would have removed some of the noise in our measurements [25]. However, it would have also meant that the results were less likely to represent a real life situation where people are free to move around to a small degree even if their occupational activities mean they are constrained to one posture. The use of an uninterrupted standing intervention, while representing some occupational settings that require continuous standing, does not represent recommendations

for the use of standing workstations, which suggest that regular changes in posture are important [10].

Additionally, the acute nature of the study does not consider the long term conditioning that may occur when this behaviour is maintained over days or weeks. However, the results of this study, when combined with others that have investigated the effect of intermittent bouts of standing, reiterate the suggestion that it may be the transitions from sitting to standing that provide the most benefit, rather than the standing itself.

## Conclusions

In healthy, normal weight participants regular activity breaks impart benefit to blood flow patterns and postprandial metabolism that are sustained for a longer duration (in the case of blood flow) and are greater magnitude (in the case of insulin metabolism) than seen with prolonged standing. Physical activity and sedentary behaviour messages should perhaps focus more on the importance of moving more, more regularly, rather than simply replacing sitting with standing.

## Supporting information

**S1 Checklist. CONSORT 2010 checklist of information to include when reporting a randomised trial**\*.
(DOC)

**S1 File. Study data set.**
(XLS)

**S2 File. Study protocol.**
(DOCX)

## Acknowledgments

We thank Karin Ongena (Department of Human Nutrition, University of Otago, Dunedin, New Zealand), who assisted with cannula insertion; Michelle Harper, Laboratory Technician (Department of Human Nutrition, University of Otago, Dunedin, New Zealand), who performed the laboratory analyses; and the participants, without whom this study would not have been possible. This research was generously supported by a grant from: University of Otago. MCP was supported by a fellowship from The National Heart Foundation of New Zealand (grant no.1518), with support from the Southland Medical Foundation.

## Author Contributions

**Conceptualization:** Meredith C. Peddie, James D. Cotter, Nancy J. Rehrer, Kate N. Thomas.

**Data curation:** Meredith C. Peddie, Kate N. Thomas.

**Formal analysis:** Meredith C. Peddie.

**Funding acquisition:** Meredith C. Peddie, James D. Cotter, Kate N. Thomas.

**Investigation:** Meredith C. Peddie, Tom Bergen, Travis D. Gibbons, Kate N. Thomas.

**Methodology:** Meredith C. Peddie, Holly A. Campbell, Kate N. Thomas.

**Project administration:** Meredith C. Peddie, Chris Kessell, Kate N. Thomas.

**Resources:** Meredith C. Peddie, Kate N. Thomas.

**Supervision:** Meredith C. Peddie, James D. Cotter, Nancy J. Rehrer, Kate N. Thomas.

**Writing – original draft:** Meredith C. Peddie.

**Writing – review & editing:** Meredith C. Peddie, Chris Kessell, Tom Bergen, Travis D. Gibbons, Holly A. Campbell, James D. Cotter, Nancy J. Rehrer, Kate N. Thomas.

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
