## [Decision Letter · Decision Letter 0]

26 Nov 2020

PONE-D-20-21865

Activity breaks but not standing improve postprandial glucose metabolism and lower-limb vascular function in an acute setting: a randomized cross-over trial

PLOS ONE

Dear Dr. Peddie,

Thank you for submitting your manuscript to PLOS ONE. After careful consideration, we feel that it has merit but does not fully meet PLOS ONE’s publication criteria as it currently stands. Therefore, we invite you to submit a revised version of the manuscript that addresses the points raised during the review process.

Both reviewers have raised concerns about data presentation and sample size calculation. In addition, Reviewer 2 underscores thinned to put in perspective the present study, as it may have important implication for public health. 

We look forward to receiving your revised manuscript.

Kind regards,

Giuseppe Andò, M.D., Ph.D.

Academic Editor

PLOS ONE

Journal Requirements:

Reviewers' comments:

Reviewer's Responses to Questions

**Comments to the Author**

1. Is the manuscript technically sound, and do the data support the conclusions?

Reviewer #1: Partly

Reviewer #2: Yes

2. Has the statistical analysis been performed appropriately and rigorously? 

Reviewer #1: I Don't Know

Reviewer #2: Yes

3. Have the authors made all data underlying the findings in their manuscript fully available?

Reviewer #1: Yes

Reviewer #2: Yes

4. Is the manuscript presented in an intelligible fashion and written in standard English?

Reviewer #1: Yes

Reviewer #2: Yes

5. Review Comments to the Author

Reviewer #1: This is an interesting report on a trial of different approaches for vascular function etc.

The title appears to report solely on a secondary outcome - this is misleading and selective reporting. The primary outcome is stated and apparently there is no significant effect. Consequently the title needs to reflect the primary hypothesis. The conclusion needs to accept that the trial stands or falls by the primary hypothesis. It needs rewording to be fair.

The analysis needs to report the overall contrast p-value before pairwise results (eg. abstract and the blood flow results) as pairwise analyses depend on the reference category.

I cannot understand the sample size calculation - FMD already appears to be a percentage, so what does a 15% difference mean here - it seems a lot larger than the actual FMD percentages, but the standard deviation and the test are missing.

There appear to be some issues with normality of data as the sd appears to be more than half the mean in some cases implying outliers that may have great leverage. What was done to allow for this? Given the small number of observations, what was done about this in the linear mixed models, as it is well known that for example ANOVA gives more significant p-values than t-tests on this sort of variable.

Given that we are looking at changes, the actual change and sd would helpfully be given. Please give effect sizes and CI wherever possible.

One assumes that treatment by period interactions were non-significant here?

In looking at percentage changes, this seems to be looked at as a linear variable, but percentage change is not normally distributed as it involves a reciprocal. It also depends very much on the size of the baseline; why are relative changes not logged?

Why for net shear rate is the estimate (-19%) not in the confidence interval (-13 to 48%)?

Is tjere a typo in Table 4 - incremental glucose AUC for regular activity - why is it ten times larger?

Reviewer #2: MS: PONE-D-20-21865 The study presented by Meredith C Peddie and coll. seems to be of interest; indeed, this question raised on the sedentary behavior, but also on the work-related constraints, could highlight a wider public health problem.

The paper is overall well written, and the study protocol seems to be well conducted.

Some question about:

Is your standard study protocol brand new or is it a protocol already used or tested, e.g. to evaluate other aspects in other studies?

About Meals: were food and drinks standardized anyway, or enrolled subjects were free to drink and eat anything? The author listed what the participants consumed, but it is unclear if each participant consumed all they listed consistently.

About Flow Mediated Dilation (FMD): authors didn’t specified the site of measurement.

About statistics: the statistical approach was not specified. Please consider the variable(s) distribution and sample size(s).

“Meals were fed at 0 and 240 min”: can the authors derive any implication –mainly on the insulin incretion – of 4-hrs fasting? Any to be corrected for in the statistics in order to evaluate the effect size of the intervention?

“This is the first study to measure the time course of the vascular and metabolic responses to Prolonged Sitting, Prolonged Standing and Regular Activity Breaks, and after consumption of a high-carbohydrate meal”. As general concept, since the authors did not mention the pulse wave velocity nowhere in the text, I think the this aspect deserve at least to be introduced.

“or the two toilet breaks were enough to mitigate the effects of sitting for long periods”: can you specify whether ALL participants had two allowed toilet breaks?

Last: I think the very strength of this study is to translate the protocol in suggestions for healthy and unhealthy people, but also for workers and employers.

Please thoroughly check your manuscript to fix any typo and grammar error.

6. PLOS authors have the option to publish the peer review history of their article (what does this mean?). If published, this will include your full peer review and any attached files.

Reviewer #1: No

Reviewer #2: No

---

## [Author Response · Author response to Decision Letter 0]

15 Dec 2020

Thank you for the opportunity to revise out manuscript. Below is a point by point response to each of the suggestions made by the editor and reviewers.

Journal Requirements:

Response: Changes have been made to the title page and to the name of the files in accordance with these templates

Response: Means and standard deviations have been added to the manuscript, and the appropriate data added to the data set which is S1_File.

Response: Supporting information file captions have been added to the end of the manuscript.

Reviewers' comments:

Reviewer #1: This is an interesting report on a trial of different approaches for vascular function etc.

The title appears to report solely on a secondary outcome - this is misleading and selective reporting. The primary outcome is stated and apparently there is no significant effect. Consequently the title needs to reflect the primary hypothesis. The conclusion needs to accept that the trial stands or falls by the primary hypothesis. It needs rewording to be fair.

Response: The title has now been reworded to: The effects of prolonged sitting, prolonged standing, and activity breaks on vascular function, and postprandial glucose and insulin responses: A randomised crossover trial.

The analysis needs to report the overall contrast p-value before pairwise results (eg. abstract and the blood flow results) as pairwise analyses depend on the reference category.

Response: p-values have been added for overall intervention effect in the abstract and for the shear rate and muscle mass results. Overall intervention effect p values are reported in the Table 3 and 4 for other variables.

I cannot understand the sample size calculation - FMD already appears to be a percentage, so what does a 15% difference mean here - it seems a lot larger than the actual FMD percentages, but the standard deviation and the test are missing.

Response: Apologies for the confusing wording here – the sample size calculation has been reworded and now reads: A sample size of 18 participants provided 80% power to detect a 0.85 reduction in Dbase-adjusted FMD (α=0.05) assuming a sample size of 1.0, using a test of paired comparisons, and allowing three participants to complete the three interventions in each of the 6 possible intervention orders.

There appear to be some issues with normality of data as the sd appears to be more than half the mean in some cases implying outliers that may have great leverage. What was done to allow for this? Given the small number of observations, what was done about this in the linear mixed models, as it is well known that for example ANOVA gives more significant p-values than t-tests on this sort of variable.

Response: The assumption of mixed model regression is that the residuals are normally distributed, not the variables themselves. The normality of residuals was checked for all models, and the only variable that required log transformation were the insulin variables. The following has been added to the statistical analysis section to clarify this point: The assumptions of regression models were checked for every model and found to be met for all models other than insulin (in which the residuals were skewed). Log transformation addressed the non-normality to a large extent, but robust standard errors were used to allow for small deviations from normality. For consistency robust standard errors were then used in all models.

Given that we are looking at changes, the actual change and sd would helpfully be given. Please give effect sizes and CI wherever possible.

Response: Changes from baseline (with standard deviations) are reported for FMD, blood flow and diameter variables, For these, and all other variables for which there was a significant effect of intervention, differences and 95% confidence intervals between interventions are reported.

One assumes that treatment by period interactions were non-significant here?

Response: When using mixed model regression it is usual practice to check for period and order effects individually, not using an interaction term. The following has been added to the statistical analysis section to clarify the fact that order and period effects have been assessed. For all mixed models period and order affects were assessed individually and found not to have an important effect, however, for completeness order was included as a predictor in all final models. 

In looking at percentage changes, this seems to be looked at as a linear variable, but percentage change is not normally distributed as it involves a reciprocal. It also depends very much on the size of the baseline; why are relative changes not logged?

Response: Percentage differences for flow and AUC variables are presented for ease of interpretation. Analysis was run with the absolute variables, with differences between interventions converted to percentages post hoc because the units of flow and area under the curve are often difficult to interpret. This has been clarified in the statistical analysis section. For example:. For ease of interpretation, differences in flow between interventions converted to percentages post-hoc.

Why for net shear rate is the estimate (-19%) not in the confidence interval (-13 to 48%)?

Response: Apologies this is a typo which has not been corrected. The estimate is 19% not -19%

Is tjere a typo in Table 4 - incremental glucose AUC for regular activity - why is it ten times larger?

Response: Thank you for identifying this error – it has now been corrected.

Reviewer #2: MS: PONE-D-20-21865 The study presented by Meredith C Peddie and coll. seems to be of interest; indeed, this question raised on the sedentary behavior, but also on the work-related constraints, could highlight a wider public health problem.

The paper is overall well written, and the study protocol seems to be well conducted.

Some question about:

Is your standard study protocol brand new or is it a protocol already used or tested, e.g. to evaluate other aspects in other studies?

Response: Some elements of the study design are very similar to our previous work looking at the effects of activity breaks on postprandial glucose and insulin responses, however, other aspects – particularly those related to the assessment of artery hemodynamics are new.

About Meals: were food and drinks standardized anyway, or enrolled subjects were free to drink and eat anything? The author listed what the participants consumed, but it is unclear if each participant consumed all they listed consistently.

Response: The description of the meals has been reworded to make the standardisation of food and drink clearer. It now reads: All participants consumed the same breakfast (muesli, trim milk, juice, toast, margarine and honey) at baseline, and a snack (chocolate brownie) at four hours. The breakfast and snack combined provided 4697 kJ energy, 170 g carbohydrate (62% energy) and 35 g fat (28% energy). Gluten-free alternatives were provided on request (n=3) and provided 4716 kJ energy, 180 g carbohydrate and 33 g fat. All food was consumed withing 15 min by all participants, and no other food was consumed over this time. Water intake was provided ad libitum during the first session and then replicated in all subsequent sessions.

About Flow Mediated Dilation (FMD): authors didn’t specified the site of measurement.

Response: The site of measurement has been added. The first sentence of the description of measurement of FMD now reads: Flow mediated dilation was measured in the popliteal fossa following published guidelines (23) at baseline and 360 min, after the participant had rested quietly for >10 min in a left lateral recumbent position.

About statistics: the statistical approach was not specified. Please consider the variable(s) distribution and sample size(s).

Response: Mixed model regression was used to compare the effects of different interventions on each outcome of interest. The assumptions of these models were checked and found to be met (except in the case on insulin, where log transformation and the use of robust standard errors were used to account for skewed residuals) To provide clarity around these issues the statistical analysis section now reads: For all mixed models period and order affects were assessed individually and found not to have an important effect, however, for completeness order was included as a predictor in all final models. The assumptions of regression models were checked for every model and found to be met for all models other than insulin (in which the residuals were skewed). Log transformation addressed the non-normality to a large extent, but robust standard errors were used to allow for small deviations from normality. For consistency robust standard errors were then used in all models. No adjustment was made for multiple comparisons.

“Meals were fed at 0 and 240 min”: can the authors derive any implication –mainly on the insulin incretion – of 4-hrs fasting? Any to be corrected for in the statistics in order to evaluate the effect size of the intervention?

Response: The outcome of interest in this study was the effect of the three interventions on insulin AUC over the entire 6 hour period, therefore the effects of the 4 h of fasting was not investigated – particularly given this pattern of meal consumption was standardised across the three conditions. 

“This is the first study to measure the time course of the vascular and metabolic responses to Prolonged Sitting, Prolonged Standing and Regular Activity Breaks, and after consumption of a high-carbohydrate meal”. As general concept, since the authors did not mention the pulse wave velocity nowhere in the text, I think the this aspect deserve at least to be introduced.

Response: In this case we are using the term vascular response to refer to the artery hemodynamics including shear rate, flow, and diameter and endothelial function assessed via FMD. Pulse wave velocity was not measured in this study. 

“or the two toilet breaks were enough to mitigate the effects of sitting for long periods”: can you specify whether ALL participants had two allowed toilet breaks?

Response: In the methods section it is specified that all participants made the walk to the bathroom at the specified times, regardless of whether they needed to use the bathroom. To re-clarfiy this in the discussion a modification to wording has been made. It now reads: It is unclear as to why the 6-h sitting intervention did not reduce FMD. Perhaps in this young, otherwise healthy, homogeneous population, the stimulus was insufficient to evoke an impairment, or the walk to the toilet on two occasions (performed by every participant regardless of need) was enough to mitigate the effects of sitting for long periods.

Last: I think the very strength of this study is to translate the protocol in suggestions for healthy and unhealthy people, but also for workers and employers.

Response: Thank you for this thoughtful suggestion. The following paragraph has been added to the discussion: The results of the current study indicate that even healthy individuals benefit from interrupting sedentary time with brisk walking. Perhaps instead of providing standing desks to individuals who sit a lot at work, we should be redesigning the physical and psychosocial occupational landscapes to provide employees with opportunities to regular interrupt periods of prolonged sitting with short bouts of activity (31)

Please thoroughly check your manuscript to fix any typo and grammar error.

Response: Typos and errors have been corrected where identified.

---

## [Editor Report · Decision Letter 1]

18 Dec 2020

The effects of prolonged sitting, prolonged standing, and activity breaks on vascular function, and postprandial glucose and insulin responses: A randomised crossover trial.

PONE-D-20-21865R1

Dear Dr. Peddie,

We’re pleased to inform you that your manuscript has been judged scientifically suitable for publication and will be formally accepted for publication once it meets all outstanding technical requirements.

Kind regards,

Giuseppe Andò, M.D., Ph.D.

Academic Editor

PLOS ONE
---

## [Editor Report · Acceptance letter]

22 Dec 2020

PONE-D-20-21865R1 

The effects of prolonged sitting, prolonged standing, and activity breaks on vascular function, and postprandial glucose and insulin responses: A randomised crossover trial. 

Dear Dr. Peddie:

I'm pleased to inform you that your manuscript has been deemed suitable for publication in PLOS ONE. Congratulations! Your manuscript is now with our production department. 

Kind regards, 

on behalf of

Dr. Giuseppe Andò 

Academic Editor

PLOS ONE